# Immune Activated Cellular Therapy for Drug Resistant Infections: Rationale, Mechanisms, and Implications for Veterinary Medicine

**DOI:** 10.3390/vetsci9110610

**Published:** 2022-11-04

**Authors:** Lynn M. Pezzanite, Lyndah Chow, Alyssa Strumpf, Valerie Johnson, Steven W. Dow

**Affiliations:** 1Department of Clinical Sciences, College of Veterinary Medicine and Biomedical Sciences, Colorado State University, Fort Collins, CO 80523, USA; 2Department of Clinical Sciences, College of Veterinary Medicine and Biomedical Sciences, The Ohio State University, Columbus, OH 43210, USA; 3Department of Clinical Sciences, College of Veterinary Medicine and Biomedical Sciences, Michigan State University, Lansing, MI 48824, USA; 4Department of Microbiology, Immunology, and Pathology, College of Veterinary Medicine and Biomedical Sciences, Colorado State University, Fort Collins, CO 80523, USA

**Keywords:** mesenchymal, stromal, stem, cell, antimicrobial, antibiotic resistance

## Abstract

**Simple Summary:**

Mesenchymal stromal/stem cells have intrinsic antimicrobial properties, thus making them attractive as an alternative treatment strategy in chronic, drug-resistant bacterial infections. Recent evidence has suggested that these antimicrobial effects can be significantly enhanced by immune activation just prior to injection. This review examines the potential role for cellular therapies in treatment of drug resistant infections in veterinary medicine, drawing on insights across species and discussing the therapeutic potential of this approach overall in today’s veterinary patients.

**Abstract:**

Antimicrobial resistance and biofilm formation both present challenges to treatment of bacterial infections with conventional antibiotic therapy and serve as the impetus for development of improved therapeutic approaches. Mesenchymal stromal cell (MSC) therapy exerts an antimicrobial effect as demonstrated in multiple acute bacterial infection models. This effect can be enhanced by pre-conditioning the MSC with Toll or Nod-like receptor stimulation, termed activated cellular therapy (ACT). The purpose of this review is to summarize the current literature on mechanisms of antimicrobial activity of MSC with emphasis on enhanced effects through receptor agonism, and data supporting use of ACT in treatment of bacterial infections in veterinary species including dogs, cats, and horses with implications for further treatment applications. This review will advance the field’s understanding of the use of activated antimicrobial cellular therapy to treat infection, including mechanisms of action and potential therapeutic applications.

## 1. Introduction

Selection of antibiotic resistant bacteria in both human and veterinary medicine necessitates novel therapeutic approaches for successful management. Chronic infections, particularly those involving biofilms and multi-drug resistant organisms, evade most attempts at effective treatment. Recent reports by the National Institutes of Health (NIH), National Institute of Allergy and Infectious Diseases (NIAID), Centers for Disease Control and Prevention (CDC), World Health Organization (WHO), and Natural Resources Defense Council (NRDC) reflect the magnitude of the problem in healthcare [1,2,3,4,5,6,7,8]. In 2013, the CDC reported that an estimated two million people developed antibiotic-resistant infections annually, with greater than 23,000 cases resulting in death [1]. Similarly, antimicrobial resistance has been extensively recently documented in veterinary medicine, and considered one of the most important issues threatening animal health worldwide [9]. Conventional approaches to treatment of bacterial infections (i.e., the development of new antibiotics), are not able to keep pace with the increasing incidence of multi-drug resistant infections [3].

Antimicrobial cellular therapy (ACT) represents a new approach to address the growing issue of chronic, drug-resistant infection. This approach employs living cells, mesenchymal stromal or ‘stem’ cells (MSC), to augment the activity of conventional antibiotic therapy. Recent work has focused on optimizing cellular therapeutic strategies to focus on use of ACT as an adjunctive therapy for multi-drug resistant (MDR) bacterial infections, including both acute and chronic cases, as will be discussed in this review. This work builds off the use of MSC for treatment of bacterial infections, previously reported in the lung or peritoneal cavity [10,11,12,13] and particularly in biofilms [14,15,16,17,18,19,20,21,22,23,24,25,26,27] and previous work by other groups demonstrating that pre-activation of MSC with inflammatory licensing agents enhances the antibacterial and immunomodulatory abilities of MSC which may enhance their effect in treatment of infection [16,17,24,25,26,28,29,30,31,32,33,34,35,36,37,38,39,40,41,42,43,44,45,46]. Summary of the studies detailing the antimicrobial effects of mesenchymal stromal cell therapy in treatment of bacterial biofilms and that activation of MSC enhances their innate antibacterial and immunomodulatory effects are detailed in Table 1 and Table 2, respectively.

Several key features distinguish the current version of ACT from other forms of cellular therapy for treating infections. First, the use of allogeneic MSC that have been activated with toll or nod-like receptors prior to administration. Pre-activation takes advantage of receptors that are commonly present in inflammation and infection to enhance the migratory properties of MSC and activate host innate immune defenses against infection [16,17,19,24,25,26,28,29,30,32,33,34,36,37,38,39,41,44,45]. A second defining characteristic of this approach in ACT is the use of repeated cell infusions for optimal effect. In addition, both intravenous and local routes of delivery were explored [43]. Systemic administration ensures that activated MSC will reach sites of deep-seated infection via chemokine-mediated migration and interact fully with the host immune response to stimulate effective antibacterial immune responses. However, intra-articular administration in an equine model of septic arthritis demonstrated a beneficial effect in localized disease processes such as those isolated to synovial structures suggesting that route of administration may be tailored to the specific disease process [25]. Finally, the concurrent administration of conventional antibiotics with ACT enhances the effect in an additive or synergistic manner, which we will discuss further.

Evidence for the effectiveness of the ACT approach has been generated in both mouse models [17,24,36,45], pet dogs with spontaneous chronic, drug-resistant bacterial infections involving soft tissues and bones [26], and an induced case–control study modelling septic arthritis in horses [25]. Thus, there is compelling preclinical evidence that ACT may be an effective means of stimulating clearance of recalcitrant, drug-resistant infections. In this article, we will review the evidence supporting use of TLR agonism to improve cellular therapy in treatment of bacterial infections in murine, canine, and equine disease models and further discuss mechanisms of action by which ACT exerts an effect. Finally, we will discuss the implications of these studies in the clinical application of cellular therapy to manage patients with intractable MDR infections.

## 2. Principles of Cellular Therapy to Treat Bacterial Infection

### 2.1. Mechanisms of MSC Antimicrobial and Immunomodulatory Action

Direct antimicrobial activity of MSC from multiple species and tissue sources has been reported, primarily through secretion of antimicrobial peptides that potentiate the activity of conventional antibiotics by increasing drug permeability of bacterial cell walls [13,16,43,47,48,49,50,51,52,53,54,55,56,57]. In addition, while MSC themselves express low immunogenicity, MSC are immunologically active, suppressing inflammation associated with infection by both direct cell-to-cell contact and secreted factors [57,58,59,60,61,62,63] including immune suppressive cytokines (e.g., IL-10, TGF-ß), metabolites (e.g., IDO, PGE2, adenosine), and matrix factors (e.g., galectins) [19,57,62,64,65,66,67,68,69]. MSC secreted factors not only suppressed biofilm formation but further disrupted formed biofilms in vitro [23,70]. MSC embedded implants have previously been demonstrated to have enhanced bacterial clearance and be more resistant to biofilm formation [15]. As biofilms are a defining feature of chronic bacterial infections, including those involving bone, synovial structures, and implants [15,71,72,73], the biofilm dispersing properties displayed by MSC are key to their role in treatment of chronic infection. The rationale for and approach to ACT takes advantage of and optimizes these innate properties of MSC for enhanced treatment of MSC [31,42,57,74] (Figure 1).

### 2.2. Cellular Activation Techniques

The functional properties of MSCs can be modified through activation of Toll-like receptors (TLR), nucleotide-binding oligomerization domain (NOD-like receptors or NLRs), or RIG-I-like receptors (RLR) [75]. Toll-like receptors (TLRs) specifically have been recognized as regulators of stromal cell functions, including survival, differentiation, and growth [35], with thirteen different TLRs identified to date in mammalian species [35]. TLRs are expressed either on intracellular membranes of the endoplasmic reticulum, lysosomes, and endosomes (TLRs 3, 7, 8 and 9) or on the cell surface (TLRs 1, 2, 4, 5, and 6) [42]. MSC derived from multiple tissue sources and species express TLRs (e.g., TLR2, TLR3, TLR4, and TLR9), which play an important role in their regulatory effects in immune modulation and response to inflammation in infection [33,76], and signaling through TLR pathways is regulated at multiple levels from transcriptional to post-translational [42]. Furthermore, interactions between TLR pathways and micro-RNAs (miRNAs) dictate either suppression or activation of the TLR signaling and downstream responses in MSCs [42]. Differences in TLR stimuli used, culture conditions or MSC source have been shown to play a role in resultant action following MSC priming, leading to inconsistent findings reported with TLR activation of MSC [31]. MSCs activated with TLRs have been demonstrated to exhibit immunosuppressive properties through induction of indoleamine-2,3-dioxygenase-1 via protein kinase R and interferon-ß [29] and to recruit immune inflammatory cells, through upregulation of secretion of immunomodulatory cytokines (CCL5, IL1ß, IL-6, IL-8) [30]. In vivo injection of various ligands (NLR2, TLR3,4 and 5) further enhanced proliferation of MSCs, increased cloning efficiency, and affected cell differentiation [36].

Importantly, activation with different TLR ligands have resulted in differential effects [46]. For example, TLR4 activation was found to induce a pro-inflammatory phenotype in MSC, termed *MSC1*, whereas TLR3 activation resulted in an *MSC2* phenotype with upregulation of more immunosuppressive pathways [77,78,79]. TLR3 but not TLR4 primed MSC enhanced their immune-suppressive activity again natural killer cells, through modulation of natural killer group 2D ligand major histocompatibility complex class I chain A and ULBP3 and DNAM-1 ligands, which was also found to be context dependent to the site of inflammation [34]. Ligation of TLR3 and TLR4 further inhibited MSCs’ ability to suppress T-cell proliferation by affecting Notch signaling pathways, which are transmembrane receptor proteins important in cell–cell communication, solidifying MSCs’ role in immunosuppression [28,37]. In addition, TLR4 activation can stimulate the release of cytokines, especially immunomodulatory chemokines such as MCP-1 and IL-8 that recruit monocytes and neutrophils, respectively [41]. Priming of equine MSC with both TLR3 and TLR4 increased expression of CXCL10, CCL2, and IL-6 and resulted in decreased T cell proliferation (TLR3 to a greater extent than TLR4) [39]. TLR3 agonist polyinosinic:polycytidylic acid (poly I:C) stimulation of MSC further regulated key innnate immune cells known to be important to anti-viral immunity in a time-dependent fashion where early activated MSC secrete type I interferon to enhance NK cell effector function and at later time points produce greater amounts of IL-6 and TGF-ß to induce senescence in NK cells and terminate inflammatory responses [38].

Furthermore, ligation of specific TLR agonists (eg., TLR2 versus TLR4 activation) can actually inhibit MSC migration, MSC-mediated immunosuppression, and reduce expansion of regulatory T cells, diminishing MSC potential effect in treating inflammatory disease [33]. In another study, inhibition of TLR4 resulted in reduced proliferation and osteogenic differentiation of adipose derived MSC. These findings indicate that TLR receptors also regulate cell differentiation pathways, which may be relevant in the setting of bacterial infections where multiple different TLR and NLR ligands are expressed.

In a study evaluating the effect of TLR activation of murine MSC in the treatment of pulmonary infection, activation with TLR 2, 4 and 9 resulted in significantly decreased production of pro-inflammatory cytokines IL-6 and TNF- α [17]. Finally, multiple aspects of culture techniques, including time of TLR agonist exposure, concentration of TLR agonist, and MSC concentration during cell activation have all been demonstrated to affect both the immunosuppressive and the antibacterial activity of MSC [24,44]. These studies provide some explanation for the previously conflicting reports regarding overall net effects of TLR stimulation, suggesting MSC polarization and ligand selection are important aspects to consider in application of TLR agonists to activation of MSC in clinical scenarios. Specifically, MSC polarization refers to the process by which MSCs may be polarized by downstream TLR signaling into two relatively homogeneous phenotypes previously classified as MSC1 and MSC2, providing both a mechanism by which to reduce heterogeneity in cellular populations and potentially improve efficacy of current cell-based therapies [77]. Taken together, these findings support the concept that MSCs’ immunomodulatory and antimicrobial function can be significantly upregulated just prior to injection by priming or ‘licensing’ with innate immune ligands such as TLR agonists, and that selection of these agonists can significantly impact the quality and the magnitude of the downstream pathways that are activated.

Activation of MSC with TLR ligands stimulates production of antimicrobial peptides, including lipocalin-2, hepcidin, and beta-defensin-2, and cathelicidin [11,32,48,51,80]. Stimulation of MSC with IFN- γ, as would typically be found in an inflammatory microenvironment as in bacterial infection, resulted in enhanced mRNA expression of TLR3 as well as IDO1, and increased secretion of immunomodulatory cytokines including IL-10 [81]. When Toll-like receptor (TLR) activation was compared to that of nucleotide-binding oligomerization domain (NOD)-like receptor (NLR) ligand stimulation of MSC specifically to enhance antimicrobial properties and immunomodulation, activation with TLR3 ligand poly I:C increased bactericidal activity, suppressed biofilm formation, enhanced neutrophil bacterial phagocytosis and increased immunomodulatory cytokine secretion (MCP-1) by equine MSC compared to nonstimulated MSC and activation with other TLR and NLR agonists [24]. Of all ligands evaluated, MSCs treated with TLR3 ligand poly I:C, of all ligands evaluated, resulted in greater production of indoleamine 2,3-dioxygenase (IDO), a clinically relevant therapeutic factor, and attenuated pathology in a mouse model of dextran sodium sulfate (DSS) induced colitis [82]. In an additional in vivo mouse model of chronic wound infection, mice treated with TLR3 activated MSC demonstrated migration to the site of infection, which was mechanistically shown to be mediated in part by upregulation of CXCR4 expression [16]. For example, activated MSC migrated more efficiently to an SDF-1 stimulus in vitro, and to sites of wound infection in vivo. Thus, pre-activation with a TLR ligand such as pIC was demonstrated to augment MSC antimicrobial activity through a variety of indirect mechanisms and was moved forward in clinical studies in dogs with naturally occurring wounds and horses with septic arthritis involving multidrug resistant organisms.

### 2.3. Route of Administration, Dosing, and Number of Injections

Both systemic and local intraperitoneal or intrasynovial injection of MSC have resulted in successful treatment of infection in animal models [25,82,83] and supports previous studies demonstrating that priming of MSC induces population-normalizing effects that can standardize what would otherwise be heterogenous cell populations [83]. Doses of 2 × 10^6^ cells/kg and up to 1 × 10^9^ cell/kg, which have previously been reported as optimal for immunomodulation in humans and large animals [84], were injected intravenously in mice with chronic *Staphylococcus aureus* impregnated implant infections and dogs with chronic naturally occurring wounds [16]. Mechanistically, when administered systemically via intravenous administration, MSC have been shown to interact with host innate immune cells, principally neutrophils and monocytes, at multiple sites, including lungs, spleen, liver, and sites of infection [64,65,85]. For example, these effects resulted in enhanced bacterial phagocytosis, mediated by MSC-secreted cytokines such as interleukin-18 (IL-8) and stimulation of neutrophil extracellular trap (NET) formation, leading to enhanced bacterial killing and neutrophil survival [16,26,78]. Recruitment of monocytes to sites of inflammation, such as bacterial infection, is mediated by chemokine CCL2 (MCP-1) produced by MSC, which mobilizes release of inflammatory monocytes from bone marrow and recruitment to sites of high CCL2 production (i.e., infection) [34]. Once recruited to wound tissues, monocytes rapidly differentiate to macrophages; important to the mechanism of ACT, TLR-3 activated MSC induce differentiation of wound macrophages from an M1 (pro-inflammatory) to M2 (reparative) phenotype [16]. This response is consistent with the anti-inflammatory phenotype of TLR-3 activated MSC previously reported [77,78,79].

When ACT was further explored in a large animal model of septic arthritis, local administration was investigated to minimize the need for larger numbers of MSC when dose was extrapolated to increased body mass [25], with positive results in reduction of local and systemic inflammation, decreased bacterial burden within joints and improved pain scores [25]. Furthermore, in a mouse model of induced colitis, intraperitoneal but not intravenous injection of TLR3 activated MSC was found to attenuate disease severity [82]. In previous studies, local injection of MSC at sites of wound infection have not been appreciated to be as effective as systemic administration [16], indicating that further investigation and comparison of routes of administration is warranted and the optimal route for a particular clinical scenario may depend on a number of factors. These studies illustrate the pros and cons of different routes of administration depending on the size of the patient, cost considerations, and condition for and accessibility of the lesion for which MSC are being administered.

Multiple versus single administrations may further improve eradication of chronic infections, theoretically due to a cumulative impact on activation of host defenses [25]. In studies performed in pet dogs with chronic MDR infections, some animals received up to 10 MSC infusions via intravenous administration [16]. A potential concern with the use of repeated injections of allogeneic MSC is the potential for induction of harmful host adaptive immune reactions to infused MSC; however, no adverse events were seen in dogs or horses receiving multiple MSC administrations for chronic infections, which may reflect the high level of systemic and local inflammation already present in multidrug resistant infections [16,25]. Future studies may employ recently investigated techniques to reduce immunogenicity when injecting allogeneic MSC such as major histocompatibility (MHC) haplotyping and matching or TGFß2 stimulation to reduce immunogenicity to MSC-mismatched stromal cell donors [86,87]. (Table 1 and Table 2).

### 2.4. Combination of MSC with Antibiotics for Enhanced Bacterial Killing

Co-administration of antibiotics with activated MSC has been a key feature of ACT for optimal bactericidal effect. Based on our studies, all major classes of antibiotics including beta-lactam drugs (penicillins, cephalosporins, carbapenems), aminoglycosides, fluoroquinolones, glycopeptide (vancomycin), and cyclic lipopeptide (daptomycin) antibiotics exhibit synergistic or additive activity with MSC secreted factors in vitro [70]. In support of this concept, the most effective treatment protocol for mice with chronic biofilm infections was activated MSC in combination with antibiotics compared to antibiotics alone, or activated or non-activated MSC alone [16]. Furthermore, canine clinical studies with spontaneous MDR infections demonstrated that administration of antibiotics to which the infecting bacteria are resistant can still be combined effectively with activated MSC treatment.

## 3. Evidence for Antimicrobial Activity in Animal Models

### 3.1. Rodent Models of Infection

Multiple rodent studies have supported both the antimicrobial effects of MSC in treatment of infection at various sites (e.g., thoracic and peritoneal cavities, subcutaneous chronic implant) [17] as well as the benefits of priming of MSC in culture prior to administration [17]. Mice with *Streptococcus pneumoniae* pulmonary infection treated with MSC exhibited reduced myeloperoxidase activity in the lungs, decreased neutrophil number in bronchoalveolar lavage fluid and lower levels of pro-inflammatory cytokines as well as bacterial load in the lungs following treatment [17]. In this model, activation of the murine MSC with TLR agonists 2,4,9 or live *S. pneumoniae* bacteria resulted in reduced production of IL-6 and TNF- α [17]. Intraperitoneal administration of TLR3 polyI:C activated MSC further reduced disease severity in mice with DSS-induced colitis through enhanced immunosuppressive activity by stimulating MSCs to increase production of indoleamine 2,3-dioxygenase (IDO) [82]. MSC can also be combined with various substrates or polymers to increase immune modulation ability [88]. In an acute model of bacterial wound infection, Kudinov et al. demonstrated that the combination of proteins secreted from MSC along with chitosan gel was able to ameliorate the presence of microorganisms in the burn wound area [89].

### 3.2. Naturally Occurring Canine Model of Chronic Infection

Dogs represent a translational model for orthopedic implant infection in humans as they develop naturally occurring implant infections in similar body sites which involve similar bacterial pathogens and antibiotic resistance patterns as chronic infections in humans. As infections were naturally occurring, induction in laboratory species could be avoided. Therefore, using the dog as a realistic, translational chronic infection model, activated allogeneic MSC were administered repeatedly intravenously without negative side effects, and in many cases, resolved infections that had resisted prolonged treatment (i.e., weeks to months) with conventional antibiotics. The canine model also addresses key issues regarding the scalability of ACT for treatment of chronic infection, as dogs in these studies have been treated with comparable doses of activated MSC (typically 2 × 10^6^ cells per kg body weight) that have also been used for systemic MSC infusion in humans [35,90]. Moreover, dogs as an outbred species also address the safety issue of repeated intravenous delivery of fully allogeneic MSC, as the donor source for MSC in all the dog studies reported by our group were adipose tissues of unrelated dogs [16]. Adverse events associated with multiple repeated infusions of activated canine allogeneic MSC over periods of up to six months were not observed, and clinical study animals have now been followed for at least two years with no subsequent adverse events noted.

### 3.3. Induced Equine Model of Septic Arthritis

The encouraging findings demonstrated with TLR activation of MSC in vitro and in murine and canine models of infection prompted further evaluation of ACT in a large animal (equine) model of septic arthritis. The equine preclinical model is a clinically and translationally relevant model for human infection for several reasons. Development of infectious arthritis as a naturally occurring disease process in horses is well-documented, their large joint volume allows for repeated collection of synovial fluid to analyze a larger number of outcome parameters and their cartilage thickness, joint volume and loading forces more closely replicates that of people than many other veterinary species [91,92,93,94,95,96]. In this work, multi-drug resistant *Staphylococcal* septic arthritis was treated with three intra-articular injections of TLR3-activated MSC and antibiotics or antibiotics alone. Horse pain scores, diagnostic imaging findings (ultrasound, magnetic resonance imaging), quantitative bacterial counts, systemic parameters of inflammation (neutrophil counts and acute phase marker serum amyloid A), and intra-synovial cytokine levels of pro-inflammatory cytokines interleukin-6 and interleukin-18 were improved in MSC + antibiotic treated horses and no adverse events were noted (Figure 2). These studies serve as strong evidence that the use of ACT has considerable promise as a new approach to management of chronic and/or multidrug resistant infections.

## 4. Discussion

Cellular therapy is emerging as a promising adjunctive therapy to combat the growing problem of drug-resistant bacterial infections and those involving biofilms, and investigation of strategies to improve potency of MSCs in an ongoing area of research [42]. While there remains an incomplete understanding of the underlying mechanisms of action of TLR agonism in ACT, as well as the demonstrated additive and synergistic effects with specific antibiotics, it is apparent from these studies that TLR-activated cellular therapy for treatment of infection is well-tolerated, effective, and can be readily implemented using allogeneic sources (i.e., bone marrow or adipose tissue derived MSC obtained from young, healthy, unrelated donors) and in a variety of chronic inflammatory disease states [74]. The site of infection also does not appear to be a limiting factor, as intravenous delivery of cells was sufficient to home to sites of infection in mice and dog models and intrasynovial injection was used to effectively treat localized infections in horses. Moreover, specific resistance patterns or bacterial strains do not seem to reduce the antimicrobial effect of MSC, as activity of ACT has been observed against a variety of different Gram-positive and -negative bacterial isolates, many displaying multiple antibiotic resistances and for which development of resistance is very different. Further characterization of the effect of TLRs in biological regulation of stromal cell function could improve MSC-based cellular immunotherapies in treatment of infection [74].

Despite promising pre-clinical studies, potential obstacles to clinical implementation of ACT still must be addressed. Regulatory pathways for approval of veterinary cellular therapies in the United States by the Food and Drug Administration (FDA) is a lengthy and expensive process, with none approved to date despite greater than ten years of development efforts. Furthermore, the primary target for the majority of cellular therapies is osteoarthritis, as the market for infections in veterinary medicine may not justify development costs. In addition, there is generally a lack of spontaneous animal models of chronic infection in which to evaluate activated cellular therapies and therefore to use for FDA approval. Finally, the use of cellular therapy specifically to treat chronic drug resistant infections was not reported until 2017 by Johnson et al., so therapy for this specific indication is relataively early in the development process. As a result, a more complete understanding of the mechanisms of action of cellular activation and optimal combinations with various antibiotics is indicated. Recent evidence suggests that long noncoding RNAs (lncRNAs) regulate a wide range of biological processes and are differentially expressed in TLR3 activated MSC, providing some framework for better understanding the molecular mechanisms by which TLR activation modulates MSCs’ functions [35]. Another potential issue is donor-to-donor MSC variability as MSCs from different genetic backgrounds have been shown to exhibit distinct antibacterial phenotypes [83], which at present has been addressed by using MSC derived from young, healthy donor animals and avoiding extensive MSC passaging. Hirakawa et al. recently demonstrated that CRISPR-based gene modulation could be used to engineer MSCs with enhanced antibacterial properties through upregulation of CD14, and further investigation of these methods is indicated [83]. The relative impact of the host immune status on response to ACT is also a potential treatment variable, which may limit improvement following ACT therapy in elderly or immunocompromised patients. The optimal number of ACT treatments has also not been established, nor is it clear which clinical parameters (i.e., biomarkers) are best suited to monitor treatment responses, or time frame at which to assess treatment impact as response may take weeks to months to manifest in the case of persistent, chronic bacterial infections. Finally, recent studies have begun to investigate the application of MSC derived exosomes as an acellular therapy capable of reparation [97], immunomodulation and drug-delivery, specifically in the context of treating sepsis, which may represent a promising future direction for anti-infective cellular therapies.

## 5. Conclusions

In summary, the use of activated cellular therapy to manage refractory or drug resistant bacterial infections is promising as an innovative option to augment antibiotic therapy. Further evaluation of mechanisms of action and investigation of ACT in randomized controlled clinical trials is indicated.

## 6. Patents

Provisional patents have been filed covering immune activated MSC technology described herein (S.D., L.P., L.C.).

## Figures and Tables

**Figure 1 vetsci-09-00610-f001:**
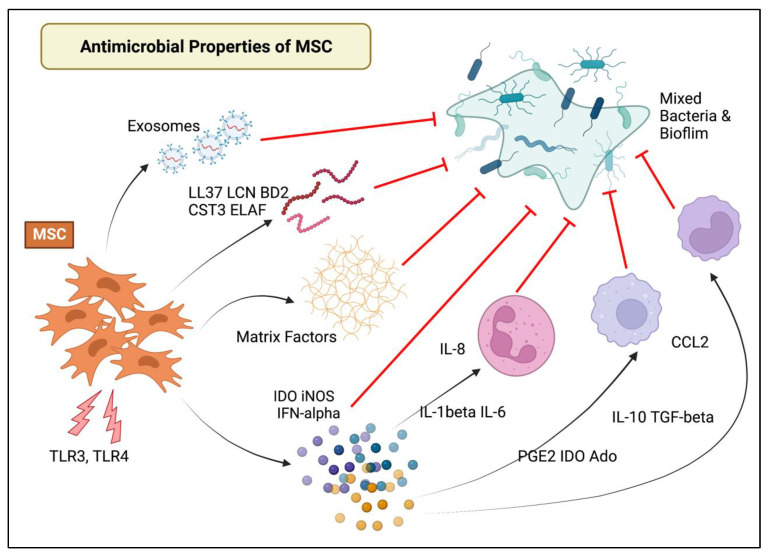
Immune mechanisms for antimicrobial properties of MSC against biofilms. Direct antimicrobial activity of MSC via secreted factors including antimicrobial peptides and indirect immunomodulatory activity of MSC are illustrated. Directly, cationic antimicrobial peptides (e.g., cathelicidin, lipocalin-2, ß-defensin 2), induce damage to bacterial membranes or alter bacterial function either directly or indirectly. Indirectly, MSC activate host immune cells, modulate local inflammation and induce angiogenesis and fibrogenesis, targeting several different cell types including T cells, macrophages, neutrophils, and dendritic cells. This activity is primarily mediated by up-regulation or inhibition of immunomodulatory cytokines and chemokines that in turn augment the immune system either to a pro-inflammatory or an anti-inflammatory state.

**Figure 2 vetsci-09-00610-f002:**
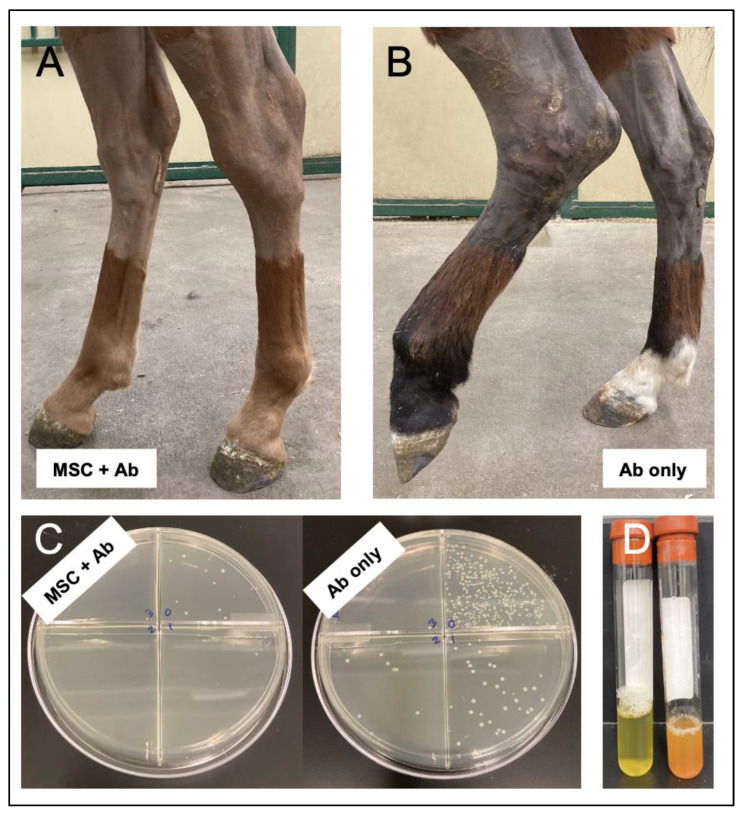
Evaluation of TLR poly I:C activated bone marrow derived MSC therapy in an equine model of multi-drug resistant USA300 methicillin resistant *Staphylococcus aureus* [25]; original unpublished images presented with permission from the authors]. Representative images of horses at day 7 following intra-articular inoculation of the left tarsocrural joint treated with three intra-articular injections of (**A**) MSC and antibiotics, or (**B**) antibiotics alone. Quantitative bacterial cultures were significantly reduced in horses treated with (**C**) MSC and antibiotics versus (**D**) antibiotics alone. Synovial fluid parameters serum amyloid A, lactate, and inflammatory biomarkers IL-6 and IL-18 were significantly improved in horses treated with MSC and antibiotics (left) versus antibiotics alone (right) (**D**).

**Table 1 vetsci-09-00610-t001:** Summary of studies demonstrating efficacy of (MSC) and conditioned medium (MSC-CM) in treatment of bacterial biofilms.

Investigator	Reference	Species	Culture Conditions or Lesion	Cell Source	Cell Dose	Protocol	Route of Administration	Outcome Parameters	Main Findings
Yuan et al. (2014)	[14]	Rat	Subcutaneous infection MRSA	Bone marrow	2 × 10^7^, 2 × 10^6^,	Dosed daily for 4 doses	Intravenous	Quantitative cultures	MSC reduced bacterial colonies.
					or 2 × 10^5^ cells/rat			Immunoassays cytokines	MSC reduced cytokine expression (IL1ℬ, IL6, IL10, CCL5).
Criman et al. (2016)	[15]	Rat	Subcutaneous *E.coli*	Bone marrow	7.5 × 10^5^ MSC/mesh	MSC seeded meshes	Seeded in meshes	Microbiologic mesh evaluation	Augmentation of bioprosthetic materials with MSC enhanced
			inoculated meshes			vs non-seeded meshes		Histologic mesh evaluation	resistance to bacterial infection.
Johnson et al. (2017)	[16]	Murine	*Staphylococcus aureus*	Adipose	1 × 10^6^ cells/injection	TLR-3 poly I:C activated or not	Intravenous	IVIS luminescence imaging	Activated MSC co-administered with antibiotics was most
			implant infection model			with or without antibiotics		to determine bacterial burden	effective to reduce bacterial bioburden.
						Dosed every 3 days, 3 doses		Wound tissue histology	
		Canine	Naturally occurring wounds	Adipose	2 × 10^6^ cells/kg	TLR-3 poly I:C activated + antibiotics	Intravenous	Quantitative cultures	Repeated MSC injection resulted in clearance of bacteria
						Dosed every 2 weeks, 3 doses		Clinical signs	and wound healing.
								Phone follow-up	
Asami et al. (2018)	[17]	Murine	*Streptococcus pneumoniae*	Bone marrow	1 × 10^6^ cells/injection	Once1 hour after bacterial inoculation	Intravenous	Bacteria bronchoalveolar lavage	MSC-CM modulates TNFα, IL-6, IL-10 after
			pulmonary infection					Myeloperoxidase activity assay	stimulation with TLR2, TLR4, TLR9 ligands.
								Bichinchoninic acid protein assay	MSC-CM suppresses CXCL1, CXCL2 production
								Histopathologic examination	after stimulation with TLR2 and TLR9 ligands.
									MSC IV decreased total cells, neutrophils, and
									myeloperoxidase activity during pulmonary infection.
									MSC IV decreased BALF cytokine levels TNFα, IL-6,
									IFN-γ, CCL2, GM-CSF during pulmonary infection.
Wood et al. (2018)	[18]	Human	In vitro *Staphylococcus aureus,*	Adipose	N/A		In vitro	Scanning electron microscopy	MSC inhibited *P. aeruginosa* biofilm formation
			Pseudomonas co-culture					Colony forming units	due to bacterial adhesion, engulfment/phagocytosis
								Biofilm assay	and secretion of antibacterial factors.
Chow et al. (2019)	[19]	Human	*Staphylococcus aureus*						
			In vitro biofilm assay	Bone marrow	N/A	TLR and Nod-like receptor agonists	In vitro	Live/dead biofilms confocal microscopy	MSC secreted factors disrupted *MRSA* biofilm formation.
			Mouse mesh implant model		1 × 10^6^ cells/injection	TLR-3 poly I:C activated with antibiotics	Intravenous	bacterial density via IVIS live imaging	Activated MSC treatment decreases bacterial bioburden
						dosed every 3 days for 4 doses			in mouse chronic biofilm infection model.
Bujnakova et al. (2020)	[20]	Canine	In vitro biofilm	Bone marrow	N/A	In vitro coculture *S. aureus, E.coli* biofilms	In vitro	Disc diffusion test	MSC-CM inhibited biofilm formation and quorum sensing.
			*Staphylococcus aureus*					Spectrophotometric crystal violet assay	
			*Escherichia coli*					Bioluminescence assay	
Bahroudi et al. (2020)	[21]	Human	In vitro *Vibrio cholerae*	Bone marrow	N/A	MSC secretome coculture	In vitro	Plate crystal violet assay	MSC secretome prevented biofilm formation
			co-culture with MSC secretome			V. cholerae 1:8 to 1:128			of *Vibrio cholerae* in a dose-dependent manner.
Marx et al. (2020)	[22]	Equine	In vitro *Pseudomonas,*	Peripheral blood	N/A	In vitro co-culture with *Pseudomonas*	In vitro	Protease array	MSC secretome inhibits biofilm formation and mature
			*Staphylococcus* biofilms			and *Staphylococcus* biofilms		Confocal microscopy biofilm composition	biofilms of Pseudomonas and Staphylococcus spp.
								Western blot analysis	MSC secrete cysteine proteases that destabilize MRSA
									biofilms increasing efficacy of antibiotics.
Marx et al. (2021)	[23]	Equine	*Ex vivo* equine skin	Peripheral blood	N/A	In vitro co-culture MSC-CM	In vitro explant	Immunofluorescence activity	MSC decreased MRSA viability in mature biofilms.
			biofilm explant model			with MRSA and MSSA		Biofilm live/dead staining	Equine MSCs secrete CCL2 that increased antimicrobial
									peptide secretion by equine keratinocytes.
Pezzanite et al. (2021)	[24]	Equine	In vitro MRSA biofilm assays	Bone marrow	N/A	TLR-3, TLR-4 NOD activated MSC	In vitro biofilms	Bactericidal activity	MSC stimulation TLR3 poly I:C suppressed biofilm formation
								Neutrophil bacterial phagocytosis	enhanced neutrophil phagocytosis
								Cytokine analysis	increased MCP-1 secretion,
								Antimicrobial peptide secretion	enhanced antimicrobial peptide production.
Pezzanite et al. (2022)	[25]	Equine	In vivo MRSA septic arthritis	Bone marrow	20 × 10^6^ cells/joint	TLR-3 poly I:C activated MSC	Intra-articular	Clinical pain scoring	Activated MSC therapy resulted in improved pain scores,
								Quantitative bacterial cultures	ultrasound and MRI scoring, quantittative bacterial counts,
								Complete blood counts	systemic neutrophil and serum amyloid A,
						Dosed every 3 days for 3 doses		Cytokines synovial fluid, plasma	synovial fluid lactate and serum amyloid A
								Imaging (radiographs, ultrasound, MRI)	synovial fluid IL-6 and IL-18.
								Macroscopic joint scoring	
								Histologic changes	
Johnson et al. (2022)	[26]	Canine	Naturally occurring chronic	Adipose	2 × 10^6^ cells/kg	TLR-3 poly I:C activated with antibiotics	Intravenous	Quantitative cultures	Repeated delivery of activated allogeneic MSC resulted
			multidrug resistant infections			Dosed every 2 weeks for 3 doses		Clinical signs	in infection clearance and wound healing.
								Phone follow-up	
Yang et al. (2022)	[27]	Human	*Pseudomonas aeruginosa*	Umbilical cord	N/A	In vitro co-culture, 8 MSC concentrations	In vitro biofilms	Titration MSC concentration	Antibacterial peptides from MSC affected biofim formation
			inoculated tracheal tubes					Anti-biofilm experiment	by downregulating polysaccharide biosynthesis
								Bacterial motility assay	protein which correlated to MSC concentration.
								DNA microarray experiment	

**Table 2 vetsci-09-00610-t002:** Summary of studies demonstrating evidence that activation of MSC enhances their innate antibacterial and immunomodulatory properties.

Investigator	Reference	Species	Culture Conditions or Lesion	Cell Source	Cell Dose	Protocol	Route	Outcome Parameters	Main Findings
Liotta et al. (2008)	[28]	Human	In vitro TLR activation	Bone marrow	N/A	TLR-3 poly I:C or TLR-4 LPS activation	In vitro	Flow cytometric evaluation	BM-MSCs expressed high levels TLR3 and 4 which induce nuclear factor k-ℬ activity, IL6, IL8, CXCL10
			T-cell co-culture					MSC differentiation assays	Ligation TLR3 and TLR4 on MSCs inhibited ability of MSC to suppress T-cell proliferation without
								T-cell proliferation assays	influencing immunophenotype or differentiation potential
								ELISA cytokines/chemokines analysis	TLR-triggering was related to impaired Notch receptor signaling in T cells
								IDO activity measures	TLR3 and TLR4 expression on MSCs provide effective mechanisms to block immunosuppressive activities
								Confocal microscopy	and restore efficient T-cell response to infection such as viruses or Gram-negative bacteria
								Quantitative analysis NFK-ℬ translocation	
								RNA extraction and rtPCR	
Opitz et al. (2009)	[29]	Human	In vitro co-culture	Bone marrow	N/A	MSC T-cells in mixed leukocyte reactions	In vitro	Karyotype analysis of MSC	TLR ligation activates innate and adaptive immune response pathways to protect against pathogens
			MSC with T-cells			TLR-3 poly I:C or TLR-4 LPS activation		Flow cytometric analysis MSC	TLR expressed on human bm-MSC enhanced immunosuppressive phenotype of MSC
								Mixed leukocyte reactions	Immnunosuppression mediated by TLR was dependent on production of IDO1
								Quantitative rt-PCR	Induction of IDO1 by TLR involved autocrine interferon signaling loop which depended on protein kinase R
								Liquid chromatography	
								Western blot analysis, siRNA	
								ELISA cell culture supernatants	
Romieu-mourez et al. (2009)	[30]	Human	In vitro activation	Bone marrow	N/A	TLR-3 poly I:C or TLR-4 LPS activation	In vitro	Flow cytometric analysis	Human MSC and macrophages expressed TLR3 and TLR4 at comparable levels
			cytokines, TLR agonists					real-time RT-PCR	TLR-mediated activation of MSC resulted in production inflammatory mediators IL-1ℬ, IL-6, IL-8/CXCL8, CCL5
								Immunoblot analysis	IFN priming combined with TLR activation increases immune responses induced by Ag-presenting MSC
								Growth response to TNF-α, IFN-α, IFN-γ	TLR activation resulted in inflammatory site attracting innate immune cells
								Immune effector infiltration analysis	
								Neutrophil chemotaxis assay	
Cassatella et al. (2011)	[32]	Human	In vitro activated	Bone marrow	N/A	TLR-3 poly I:C or TLR-4 LPS activation	In vitro	Cytofluorometric analysis	TLR-3 MSC activation enhanced anti-apoptosis of neutrophils more than TLR-4
			MSC neutrophil coculture					ELISA immunoassays	TLR-3 and TLR-4 activation enhanced respiratory burst ability and CD11b expression by PMN
								Respiratory burst cytochrome C reduction	TLR-3 activation effects mediated by IL-6, IFN-ℬ and GM-CSF
									TLR-4 activation effects mediated by GM-CSF
Lei et al. (2011)	[33]	Murine	In vitro TLR activation	Bone marrow	N/A	TLR-2 or TLR-4 activation	In vitro	MSC migration	TLR2 ligation (but not TLR4) inhibited MSC migration, MSC mediated immunosuppression on allo-MLR,
								Allogeneic mixed lymphocyte reaction	and reduced MSC mediated expansion of Treg cells
								Induction Treg cell	TLR2 activation induced lower CXCL10 mRNA and protein expressions
									TLR2 and TLR4 had different effects on immunomodulatory capacity of MSC
Giuliani et al. (2014)	[34]	Human	In vitro MSC NK cell coculture	Bone marrow	N/A	TLR-3 or TLR-4 activation	In vitro	Flow cytometry CD107 degranulation	TLR primed MSC are more resistant than unprimed MSC to IL-2 activated NK-induced killing
				Embryonic		NK cell MSC coculture		ELISA culture supernatants	TLR-primed MSC modulated naturall killer group 2D ligands MHC class I chain A, ULBP3, DNAM-1 ligands
								Chromium release assay	MSC adapt their immunobehavior in inflammatory context, decreasing susceptibility to NK killing
									TLR3 but not TL4 primed MSC enhance suppressive functionns against NK cells
Johnson et al. (2017)	[16]	Murine	*Staphylococcus aureus*	Adipose	1 × 10^6^ cells/	TLR-3 poly I:C activation +/- antibiotics	Intravenous	Bacterial burden IVIS imaging	Activated MSC co-administered with antibiotics was most effective to reduce bacterial bioburden
			implant infection model		/injection	dosed every 3 days for 3 doses		Wound tissue histology	
		Canine	Naturally occurring wounds	Adipose	2 × 10^6^ cells/kg	TLR-3 poly I:C activated with antibiotics	Intravenous	Quantitative cultures	Clearance of bacteria and wound healing following repeated IV injection
						dosed every 2 weeks for 3 doses		Clinical signs, Phone follow-up	
Gorskaya et al. (2017)	[36]	Murine	Intraperitoneal injection	Bone marrow	NLR/TLR ligands	NLR2 and TLR (LPS, flagellin, CpG, poly I:C)	Intraperitoneal	Efficiency bone marrow MSC colony formation	NLR, TLR and *S. typhimurium* antigenic complex increase efficiency of MSC cloning and content by 1 hr
			NLR, TLR, *S. typhimurium*		10 µg/mouse	and *S. typhimurium* antigenic complex			
Rashedi et al. (2017)	[37]	Human	In vitro activation TLR ligands	Bone marrow	N/A	TLR-3, TLR-4 effect on MSC Treg induction	In vitro	MSC, CD4+ lymphocyte co-culture assays	TLR3/4 activation MSC enhanced Treg generation in CD4+ lymphocyte/MSC cultures
								Gene and protein expression analysis	TLR3/4 activation augmented Treg induction via Notch pathway
								Flow cytometric analysis	
								Quantification cytokines culture medium	
Petri et al. (2017)	[38]	Human	In vitro coculture TLR-3	Nasal mucosa	N/A	TLR-3 activated MSC effect on NK cells	In vitro	ELISA immunoassays	Early time points TLR3-activated MSC secrete type I interferon to enhance NK cell effector function
			TLR-3 activated					Flow cytometric analysis	Later time points NK cell function limited by TGF-ℬ and IL-6
			MSCs and NK cells					Surface/intracellular staining	Feedback regulatory NK cells to MSCs promote survival, proliferation, pro-angiogenic properties
								Cytotoxicity assays	
								Degranulation assays	
								NK cell proliferation assays	
								MSC invasion and proliferation assays	
Cassano et al. (2018)	[39]	Equine	In vitro co-culture TLR ligands	Bone marrow	N/A	TLR-3 or TLR-4 activation	In vitro	T-cell proliferation via flow cytometry	TLR3/4 priming increased MSC expression IL6, CCL2, CXCL10
						MSC co-culture inflammatory macrophages		Macrophage RNA gene expression	TLR3/4 priming or exposure to inflammatory macrophages enhanced immunomodulatory function
						Suppression T-cell proliiferation assay			demonstrated by decreased T-cell proliferation
Cortes-Araya et al. (2018)	[41]	Equine	In vitro comparison MSC tissue sources	Endometrium	N/A	TLR-4 primed MSC versus unprimed	In vitro	Antimicrobial peptide immunocytochemistry	Lipocalin-2 was expressed at higher levels in EM-MSC than AD or BMD
			In vitro activation with TLR4 ligand	Adipose				Cytokine secretion via ELISA	TLR-4 stimulated lipocalin-2 production by all three cell types
				Bone marrow				Gene expression analyses	TLR-4 induced expression IL-6, IL-8, MCP-1, chemokine ligand-5, TLR4 by all three cell types
Asami et al. (2018)	[17]	Murine	In vitro activation with TLR ligands	Bone marrow	1 × 10^6^ cells	1 injection 1 hour after bacterial inoculation	Intravenous	Bacteria bronchoalveolar lavage	MSC-CM modulates TNFα, IL-6, IL-10 after
			*Streptococcus pneumoniae*		/injection			Myeloperoxidase activity assay	stimulation with TLR2, TLR4, TLR9 ligands.
			pulmonary infection					Bichinchoninic acid protein assay	MSC-CM suppresses CXCL1, CXCL2 production
								Histopathologic examination	after stimulation with TLR2 and TLR9 ligands.
									MSC IV decreased total cells, neutrophils, and
									myeloperoxidase activity during pulmonary infection.
									MSC IV decreased BALF cytokine levels TNFα, IL-6,
									IFN-γ, CCL2, GM-CSF during pulmonary infection.
Chow et al. (2019)	[19]	Human	In vitro *Staphylococcus aureus* biofilm assay	Bone marrow	N/A	Comparison TLR, NLR receptor agonists	In vitro	Live/dead biofilms via confocal microscopy	MSC secreted factors disrupted *MRSA* biofilm formation
			Mice with mesh implant biofilm animal model		1 × 10^6^ cells	TLR-3 poly I:C activated with antibiotics	Intravenous	bacterial density by IVIS live imaging	Activated MSC treatment decreases bacterial bioburden in mouse chronic biofilm infection model
					/injection	Dosed every 3 days for 4 doses			
Kurte et al. (2020)	[44]	Murine	In vitro splenocyte and MSC and Tcell	Bone marrow	N/A		In vitro	Quantitative real-time PCR	Time dependent LPS activation regulate IL6 and iNOS expression in MSCs.
			and MSC co-cultures				Subcutaneous	Flow cytometry	Immunosuppressive activity of MSCs on T cell proliferation depends on time dependent LPS activation.
			Murine autoimmune encephalomyelitis (EAE)					Immunosuppression assay	Long exposure to LPS enhances MSC therapeutic potential in EAE.
								Treg, Th17, Th1 differentiation assay	TLR4 expression involved in immunosuppressive capacity of MSCs in vitro.
								Thelper analysis in treated mouse lymph nodes	TLR4 inhibition disrupts capacity of MSCs to inhibit Th1 and Th17 cells in vitro.
									TLR4 deficiency reduces therapeutic effect of MSCs in EAE.
Aqdas et al. (2021)	[45]	Murine	In vitro co-culture MSC with	Bone marrow	N/A	TLR-4 or NOD-2 activated MSC	In vitro	Cytokine secretion ELISA (IL-6, IL-10, IL-12, TNF-α)	TLR4/NOD-2 augmented pro-inflammatory cytokine secretion.
			Mycobacterium tuberculosis (Mtb)					RT-qPCR (IL-6, IL-12, IL-10, iNOS, TNF-α, TGF-ℬ)	TLR4/NOD-2 co-localized Mtb in lysosomes.
								Phenotypic charactization of MSC markers	TLR4-NOD-2 induced autophagy.
								Evaluation MSC differentiation	TLR4-NOD-2 enhanced NF-κℬ activity via p38 MAPK.
								Bacterial load determination post-infection	TLR4-NOD-2 reduced intracellular Mtb survival.
								Bacterial tracking into autolysosomes	Triggering TLR4-NOD-2 pathway may be future immunotherapy.
Pezzanite et al. (2021)	[24]	Equine	In vitro MRSA biofilm assays	Bone marrow	N/A	TLR-3, TLR-4 and NOD activated MSC	In vitro	Bactericidal activity	MSC stimulation with TLR3 poly I:C suppressed biofilm formation, enhanced neutrophil phagocytosis,
								Neutrophil bacterial phagocytosis	increased MCP-1 secretion and enhanced antimicrobial peptide cathetlicidin production
								Cytokine analysis	
								Antimicrobial peptide secretion	
Johnson et al. (2022)	[26]	Canine	Naturally occurring chronic	Adipose	2 × 10^6^ cells/kg	TLR-3 poly I:C activated with antibiotics	Intravenous	Quantitative cultures	Repeated delivery of activated allogeneic MSC resulted in infection clearance and wound healing
			multidrug resistant infections			dosed every 2 weeks for 3 doses		Clinical signs, Phone follow-up	
Pezzanite et al. (2022)	[25]	Equine	MRSA inoculated septic arthritis	Bone marrow	20 × 10^6^ cells/joint	TLR-3 poly I:C activated MSC	Intra-articular	Clinical pain scoring	Activated MSC therapy resulted in improved pain scores, ultrasound and MRI scoring, quantitative
						dosed every 3 days for 3 doses		Quantitative bacterial cultures	bacterial counts, systemic neutrophil and serum amyloid A, and synovial fluid lactate, serum
								Complete blood counts	
								Cytokine analyses (blood, synovial fluid)	
								Imaging (radiographs, ultrasound, MRI)	

## Data Availability

Not applicable.

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
