# Peer review of "Immune Activated Cellular Therapy for Drug Resistant Infections: Rationale, Mechanisms, and Implications for Veterinary Medicine"

_vetsci, 2022, doi:10.3390/vetsci9110610_

Round 1

Reviewer 1 Report

Activated Cellular Therapy can be a promising additional therapeutic option in MDR infections, although the first initiatives are already 10 years old. I do miss this perspective in this paper, or at least in the discussion. Why does it take so long, what are the encountered disadvantages during those 10 years? Instead of repeating that no side effects are noted and therapy is well tolerated.

Specific remarks: 

Figure 1 line 143-144 Indirect ... of MSC.  is meant as header?

L 157 Notch signaling needs clarification

L 158 reference (rashedi, liotta) official referencing

L 162 decreased T cell proliferation: implication of ( TLR3>TLR4)? TLR3 more decreased or TLR3 more remaining?

L 162 official referencing for for (cassano 2018)

L 163 please clarify TLR3 polyI:C

L 163-164 ...regulated key innnate immune cells in important to anti-viral immunity...:  sentence not clear

L 176 redundant comma after 4

L 182 What is MSC polarization?

L 183 aspects 

L 189 stimulates enhanced production: enhanced is redundant

L 194-195 Strange introduction for abbreviation of TLR additional to NLR

L 198 description of lab circumstances (for two hours at a density in media) not clear to me

L 200-201 not clear what is meant by ...compared to resting MSC and other agonists...

L 201 polyI:C again

L 202 clarification of IDO needed

L 205 enhanced seems incorrect because mice is subject: suggestion: add demonstrate(d)?

L 219 what is "mice implant infections"?

L 256 infusions by intravenous administration, not injection (infusion implies longer duration, by slower rate than injection); suggestion to replace all injections by administrations L 257-L 262

L 269 vancomycin is NOT an example for macrolides, it belongs to the class of glycopeptide antibiotics, and

daptomycin is NOT a glycopeptide but a cyclic lipopeptide antibiotic

L 273 starting with furthermore, a verb is missing (is demonstrated?); what is the added value of using antibiotics for which resistance is shown?

L 288 unclear wording: demonstrated to be through 

L 300 delivery: better replaced by administration

L 321 reference [91=96]?

Figure 2: Please acknowledge that the figures are replicated with consent of the author (being the first author from this paper as well) and editor of [25]. It is inconvenient that this publication is not available yet. 

L378 unclear sentence (suggests and regulated seem to be conflicting)

Author Response

Reviewer 1

Activated Cellular Therapy can be a promising additional therapeutic option in MDR infections, although the first initiatives are already 10 years old. I do miss this perspective in this paper, or at least in the discussion. Why does it take so long, what are the encountered disadvantages during those 10 years? Instead of repeating that no side effects are noted and therapy is well tolerated.

  • Thank you to the reviewer for their constructive comments and for bringing up these points regarding the challenges of integration of activated cellular therapy in veterinary practice. Further elaboration on these topics has been added to the discussion section of the paper. Briefly, the regulatory pathway for approval of veterinary cellular therapies in the United States by the FDA is a lengthy and expensive process, with none approved to date despite greater than ten years of development efforts. Furthermore, the primary target for cellular therapies is osteoarthritis, as the market for infections in veterinary medicine may not justify development costs. In addition, there is generally a lack of spontaneous animal models of chronic infection in which to evaluate activated cellular therapies and therefore to use for FDA approval. Finally, the use of cellular therapy specifically to treat chronic drug resistant infections was not reported until 2017 (Johnson et al.), so therapy for this specific indication has not been in development for ten years. Thank you for requesting this information, which has been added to the discussion section.

Specific remarks: 

Figure 1 line 143-144 Indirect ... of MSC.  is meant as header?

  • Thank you for requesting this clarification – the figure legend as been edited accordingly.

L 157 Notch signaling needs clarification

  • Further information clarifying this statement and the role of Notch receptor pathways has been added.

L 158 reference (rashedi, liotta) official referencing

  • Thank you for catching this. Appropriate numbers have been added.

L 162 decreased T cell proliferation: implication of ( TLR3>TLR4)? TLR3 more decreased or TLR3 more remaining?

  • TLR3 decreased T cell proliferation to a greater extent than TLR4. This has been clarified.

L 162 official referencing for for (cassano 2018)

  • The appropriate number for this reference has been added.

L 163 please clarify TLR3 polyI:C

  • Further information clarifying the acronym has been added.

L 163-164 ...regulated key innnate immune cells in important to anti-viral immunity...:  sentence not clear

  • This has been further clarified.

L 176 redundant comma after 4

  • This has been removed.

L 182 What is MSC polarization?

  • Thank you for requesting this clarification. Further elaboration on the topic and a reference has been provided.

L 183 aspects 

  • This has been changed.

L 189 stimulates enhanced production: enhanced is redundant

  • Enhanced has been removed.

L 194-195 Strange introduction for abbreviation of TLR additional to NLR

  • This has been revised. Please advise if other specific changes are requested.

L 198 description of lab circumstances (for two hours at a density in media) not clear to me

  • This section has been revised for clarity and the details removed at the request of reviewer 2.

L 200-201 not clear what is meant by ...compared to resting MSC and other agonists...

  • This has been clarified.

L 201 polyI:C again

  • This acronym has now been defined and then used in the same fashion in the remainder of the manuscript. Please indicate if the reviewer is requesting that it be written out in each instance.

L 202 clarification of IDO needed

  • This abbreviation has been defined.

L 205 enhanced seems incorrect because mice is subject: suggestion: add demonstrate(d)?

  • This has been changed as requested.

L 219 what is "mice implant infections"?

  • This has been clarified.

L 256 infusions by intravenous administration, not injection (infusion implies longer duration, by slower rate than injection); suggestion to replace all injections by administrations L 257-L 262

  • This has been changed everywhere when noted. Please advise if there are specific instances where this should still be corrected.

L 269 vancomycin is NOT an example for macrolides, it belongs to the class of glycopeptide antibiotics, and

daptomycin is NOT a glycopeptide but a cyclic lipopeptide antibiotic

  • Thank you for catching this error. This has been corrected.

L 273 starting with furthermore, a verb is missing (is demonstrated?); what is the added value of using antibiotics for which resistance is shown?

  • This has been added.

L 288 unclear wording: demonstrated to be through 

  • This has been clarified.

L 300 delivery: better replaced by administration

  • This has been replaced.

L 321 reference [91=96]?

  • This has been changed to 91-96.

Figure 2: Please acknowledge that the figures are replicated with consent of the author (being the first author from this paper as well) and editor of [25]. It is inconvenient that this publication is not available yet. 

  • Thank you for requesting this information. Figure 2 represents original figures that are not published in reference 25, but as the reviewer indicated are from the same co-author group, which has been clarified in the text. We do not believe additional permissions are necessary from the editor of [25] as these are original. Reference 25 is currently in press and is anticipated to be available for review prior to acceptance of this manuscript. Please indicate if further action is necessary for inclusion of figure 2.

L378 unclear sentence (suggests and regulated seem to be conflicting)

  • This sentence has been clarified.

Reviewer 2 Report

This manuscript is a review on alternatives to antibiotics in the treatment of bacterial infections. The alternative highlighted is the activated cellular therapy.

It is a well written manuscript and interesting manuscript, though a review should not be used to highlight the own work, and thus phrases like 'This research group...' 'In our studies...' 'we have explored'...are not really appropriate (you can use that in grant writing) . Omit as a review should cover all research in the field. Try to avoid also things like 'in a study..' as the reference is the study. Just summarise and integrate in the text.

Line 197: detailed materials and methods are not interesting in such a review. If the reader wants to do the same, he can look up those details in the original publication.

Introduction

The introduction starts with an odd sentence, development of antibiotic resistant infections. First the process is not the development but the selection and second not the infection is antibiotic resistant but the bacterium. So change to "the selection of antibiotic resistant bacteria"

As to the reports on the increase of the prevalence of resistance, you should mention that all those reports are on human bacteria. Perhaps also include something on animal bacteria if available? You may also include the reduction of resistance in commensal bacteria in The Netherlands.

Line 46, when you speak about resistance is that an overall of resistance in all types of bacteria? One should be careful with that as several resistances are specific to certain bacterial species and absent in others, and thus the cited prevalence is also in function of the bacteria species included in the surveillance. I would omit that to avoid all confusion, but you can also specify better.

The sentence referring to complication of treatment of viral and parasitic infections by MDR bacteria does not seem logic, How can a MDR bacterium complicate the treatment of HIV, if the virus is not resistant? Please explain better what is meant.

Table 1 and 2 are interesting but only mentioned in the introduction and not really with an accompanying text. What is the conclusion? Can the table be summarised in a different way than only saying that therapies have been successful? references in the table should be numbers and not as they are now.

Line 82: pezzanite, think that is a reference. Adapt appropriately

Figure 1. This is dealing with the mechanism only for biofilms, include that in the title of the figure.

Line 165: think guiliani is to be a refence, adapt appropriately, also line 158, 162

chapter 2.3. Much of this is in the table 1 and 2 I guess? I did not check as the references did not allow to smoothly check. If so refer also to the table.

Line 269: macrolides (vancomycin), and glycopeptide (daptomycin), this is not correct, vancomycin is a glycopeptide and daptomycin is a lipopeptide antibiotic. adapt appropriately

The first sentence of chapter 3.2 is not clear. Guess you simply mean medical devices-related infections in dogs are similar to those in humans. Anyhow, in any animal species this is the case and not only dogs. As such the dog is one of the animal models that can be used. Moreover, there are several in vitro models, which do not cause animal suffering... would be good to have a good motivation for the use of dogs, what are the advantages above in vitro or other animal models. Alternatively you can refer to the fact that studies have been performed in dogs and explain the translational factor. 

line 313 the = should be a ,

In the discussion you state that ACT is promising for MDR related infections, though from the presented studies, I rather have the impression that it is mainly for biofilm related infections. For the latter, there are few to no therapeutic options and is very valid also. 

Line 369. Indeed it is unrelated to specific resistance patterns as the mechanism of resistance is very different. Only in case of the combination of ACT and antibiotics the resistance patterns are of importance.

Line 389, immunocompromised is not diabetes mellitus, it is way more than that. DM is only one of the causes of immunosuppression. 

The references need editing for line space, italics. An accepted manuscript should not figure there unless the paper is accompanied and the authors are sure to be able to give the refence at time of publication. There are other mistakes in the reference list that need to be corrected, too many to name, so best revise all.

Author Response

Reviewer 2

This manuscript is a review on alternatives to antibiotics in the treatment of bacterial infections. The alternative highlighted is the activated cellular therapy.

It is a well written manuscript and interesting manuscript, though a review should not be used to highlight the own work, and thus phrases like 'This research group...' 'In our studies...' 'we have explored'...are not really appropriate (you can use that in grant writing) . Omit as a review should cover all research in the field. Try to avoid also things like 'in a study..' as the reference is the study. Just summarise and integrate in the text.

  • Thank you for these comments. References referring the authors own work and comments including ‘in a study etc’ have been removed throughout.

Line 197: detailed materials and methods are not interesting in such a review. If the reader wants to do the same, he can look up those details in the original publication.

  • These details have been removed as requested.

Introduction

The introduction starts with an odd sentence, development of antibiotic resistant infections. First the process is not the development but the selection and second not the infection is antibiotic resistant but the bacterium. So change to "the selection of antibiotic resistant bacteria"

  • This has been changed.

As to the reports on the increase of the prevalence of resistance, you should mention that all those reports are on human bacteria. Perhaps also include something on animal bacteria if available? You may also include the reduction of resistance in commensal bacteria in The Netherlands.

  • Thank you for this suggestion. Further reference to antimicrobial resistance specifically documented in veterinary medicine has also been included.

Line 46, when you speak about resistance is that an overall of resistance in all types of bacteria? One should be careful with that as several resistances are specific to certain bacterial species and absent in others, and thus the cited prevalence is also in function of the bacteria species included in the surveillance. I would omit that to avoid all confusion, but you can also specify better.

  • This has been omitted.

The sentence referring to complication of treatment of viral and parasitic infections by MDR bacteria does not seem logic, How can a MDR bacterium complicate the treatment of HIV, if the virus is not resistant? Please explain better what is meant.

  • These statements have been removed to avoid confusion and detraction from the focus of the manuscript.

Table 1 and 2 are interesting but only mentioned in the introduction and not really with an accompanying text. What is the conclusion? Can the table be summarised in a different way than only saying that therapies have been successful? references in the table should be numbers and not as they are now.

  • Reference numbers are now included in the tables and further reference to them has been made in the body of the manuscript.

Line 82: pezzanite, think that is a reference. Adapt appropriately

  • This has been changed.

Figure 1. This is dealing with the mechanism only for biofilms, include that in the title of the figure.

  • This has been changed.

Line 165: think guiliani is to be a refence, adapt appropriately, also line 158, 162

  • Thank you for catching these errors. These have been corrected.

chapter 2.3. Much of this is in the table 1 and 2 I guess? I did not check as the references did not allow to smoothly check. If so refer also to the table.

  • Reference to tables 1 and 2 has been made in this section as well.

Line 269: macrolides (vancomycin), and glycopeptide (daptomycin), this is not correct, vancomycin is a glycopeptide and daptomycin is a lipopeptide antibiotic. adapt appropriately

  • This has been changed.

The first sentence of chapter 3.2 is not clear. Guess you simply mean medical devices-related infections in dogs are similar to those in humans. Anyhow, in any animal species this is the case and not only dogs. As such the dog is one of the animal models that can be used. Moreover, there are several in vitro models, which do not cause animal suffering... would be good to have a good motivation for the use of dogs, what are the advantages above in vitro or other animal models. Alternatively you can refer to the fact that studies have been performed in dogs and explain the translational factor. 

  • Thank you for requesting clarification. Dogs that had developed naturally occurring infections following implant placement were used as a naturally occurring animal model of disease with translational value in humans so as to avoid the induction of infection and subsequent morbidity in laboratory animals. This sentence has been clarified.

line 313 the = should be a ,

  • This has been changed to a dash.

In the discussion you state that ACT is promising for MDR related infections, though from the presented studies, I rather have the impression that it is mainly for biofilm related infections. For the latter, there are few to no therapeutic options and is very valid also. 

  • Thank you for this suggestion. This wording has been altered to reflect the above and the authors’ meaning.

Line 369. Indeed it is unrelated to specific resistance patterns as the mechanism of resistance is very different. Only in case of the combination of ACT and antibiotics the resistance patterns are of importance.

  • Thank you for this comment. Further reference has been made to specify that the mechanism of resistance if different. Please advise if further changes are requested.

Line 389, immunocompromised is not diabetes mellitus, it is way more than that. DM is only one of the causes of immunosuppression. 

  • Reference to diabetes mellitus has been removed to reduce confusion.

The references need editing for line space, italics. An accepted manuscript should not figure there unless the paper is accompanied and the authors are sure to be able to give the refence at time of publication. There are other mistakes in the reference list that need to be corrected, too many to name, so best revise all.

  • Thank you for these suggestions. The accepted manuscript [25] is currently in production and will be available before the publication of this manuscript. Significant revisions to references have been made throughout this section.